



# $^{10}$Be age control of glaciation in the Beartooth Mountains, USA from the latest Pleistocene through the Holocene

Aaron M. Barth[1], Elizabeth G. Ceperley[2], Claire Vavrus[2], Shaun A. Marcott[2], Jeremy D. Shakun[3], Marc W. Caffee[4,5]

[1]Department of Geology, Rowan University, Glassboro, NJ, USA

[2]Department of Geoscience, University of Wisconsin – Madison, Madison, WI, USA

[3]Department of Earth and Environmental Sciences, Boston College, Chestnut Hill, MA, USA

[4]Department of Physics and Astronomy, Purdue University, West Lafayette, IN, USA

[5]Department of Earth, Atmospheric, and Planetary Sciences, Purdue University, West Lafayette, IN, USA

*Correspondence to:* Aaron M. Barth (bartha@rowan.edu)

**Abstract.** Alpine glaciers in the western United States are often associated with late-Holocene Little Ice Age (LIA) advances. Yet, recent studies have shown many of these glacial landforms are remnants of latest-Pleistocene retreat with only the most cirque-proximal moraines preserving LIA activity. Additionally, the timing and magnitude of
glacial advances during the Neoglacial-LIA interval remains uncertain with presumed maximum extents occurring during the LIA driven by lower Northern Hemisphere insolation levels. Here we present $^{10}$Be surface exposure ages from a glacial valley in the Beartooth Mountains of Montana and Wyoming, United States. These new data constrain the presence of the glacier within 2-3 km of the cirque headwalls by the end of the Pleistocene with implications for large-scale retreat after the Last Glacial Maximum. Cirque moraines from two glaciers within the valley preserve a
late-Holocene readvance with one reaching its maximum prior to $2.1 \pm 0.2$ ka and the other $0.2 \pm 0.1$ ka. Age variability among the moraines demonstrates that not all glaciers were largest during the LIA and presents the possibility of regional climate dynamics controlling glacial mass balance.

## 1 Introduction

Glacial retreat is one of the clearest indicators of the climate system's response to recent global warming. Photographic and satellite imagery of reductions in global glacial extent from the past century demonstrate this widespread phenomenon (Bolch, 2007; Catania et al., 2018). Within the last two decades alone, the rate of ice loss from mountain glaciers has doubled (Hugonnet et al., 2021). However, analysis of glacier sensitivity to climate change is limited by the instrumental record, which only goes back decades (Braumann et al., 2020). We are therefore reliant
on geologic records of past glacial activity to determine the influence of anthropogenic warming on glacier mass balance relative to natural variability. In the early Holocene, glaciers within the western United States (U.S.) and Canada were at minimum lengths, similar to modern, with peak Northern Hemisphere (NH) summer insolation values contributing to the retreat (Solomina et al., 2015). Reactivation of glaciation after 6 ka ("Neoglacial") occurred as glaciers advanced to their greatest Holocene extent (Porter and Denton, 1967; Solomina et al., 2016, 2015). Relative
to the last glacial period, the Holocene is considerably more stable in terms of its climate variability. However, variable



timing of glacier maximum extent throughout the Neoglacial suggests other mechanisms besides NH insolation. Well-dated records of late-Holocene glaciation are therefore required to accurately assess forcing mechanisms for the Neoglacial.

Many mountain ranges of the western U.S. remained below the southern reaches of the Laurentide Ice Sheet during the Last Glacial Maximum (LGM, 26 – 19 ka; Clark et al., 2009) where numerous alpine glaciers and icecaps nucleated (Laabs et al., 2020). LGM glacial positions are recorded as down valley moraines extending as much as 50 km from the cirques while younger, high-elevation glacial landforms are preserved within 1-2 km of the headwall (Davis, 1988; Davis et al., 2009). These younger glacial deposits were originally thought to record Neoglaciation, but recent dating concluded that many of these landforms across the western U.S. are in fact remnants of latest Pleistocene

and earliest Holocene glaciation (Marcott et al., 2019). The questions then arise: what is the record of late Holocene glaciation in the western U.S., if any exists at all? What does it tell us about the response of mountain glaciers to climatic forcings throughout the Holocene, and were they sufficient to permit glacial regrowth?

Here we present [10]Be exposure ages from high-elevation glacial landforms in the Beartooth Mountains of Montana (MT) and Wyoming (WY) to determine the timing of glaciation in this sector of the western U.S. Together

with previously published ages on downvalley LGM moraines, this new chronology sheds light on the rate and magnitude of glacier retreat during the last deglaciation as well as potential Neoglacial regrowth during the late Holocene.

## 2 Geologic and geomorphic background

The Beartooth Mountain range extends from southwestern MT into northern WY and is a broadly arcuate mountain range reaching ~30 km at its widest. At its base, the Mill Creek-Stillwater Fault Zone separates the northern foothills of the Beartooths from the Great Plains (Fig. 1; Bevan, 1923; Montgomery and Lytwyn, 1984) with ~1800 m relief between the Plains (~1800 m asl) and the highest elevations in the mountain range (>3600 m asl). High plateaus have been dissected by fluvial and glacial erosion. Numerous cirques and northward-oriented glacial valleys

are present within the Beartooths, whereas the Absoroka Range to the south exhibits more south- and eastward-oriented glacial valleys. The eastern portion of the mountain range is predominantly Archean quartz-rich granitic gneisses and migmatites with inclusions of metasedimentary and metaigneous rocks (Van Gosen et al., 2000). Course pegmatitic dikes are common throughout the region (Bevan, 1923).

Glacial landforms in the mountain range record multiple phases of glaciation, including the Bull Lake

(Marine Isotope Stage (MIS) 6), and Pinedale (MIS 2) glaciations. Bull Lake glaciation within the northeast sector of the Beartooths is recorded as remnant outwash terraces of higher elevation to those of the younger Pinedale terraces (Ballard, 1976). Morainal evidence, however, is sparse for Bull Lake glacier limits and is suggestive of similar, or lesser, glacial extents to Pinedale advances (Licciardi and Pierce, 2018). During the last glaciation, high-elevation ice flowed southward to the Yellowstone Plateau where it coalesced with similar glaciers from the surrounding mountains

forming the Yellowstone ice cap, which acted as an ice divide at the LGM (Licciardi and Pierce, 2018, 2008). Numerous, long (>20 km) glacial valleys are present along the periphery of the Greater Yellowstone Glacial System (GYGS), including those of the northern Beartooths. Glacial deposits including erratics, moraines, and outwash

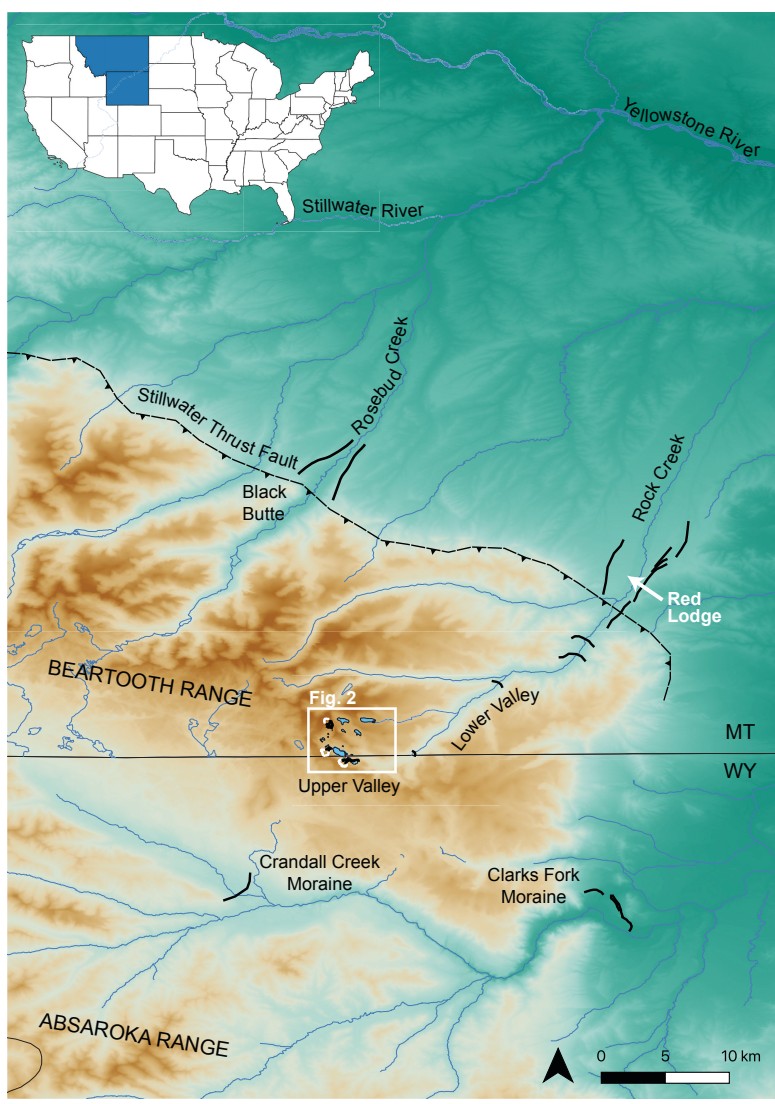

**Figure 1 - Regional Map -** Map of the Beartooth Mountain range in Montana and Wyoming, United States. Brown colors indicate higher elevation, and green colors lower elevation. Solid black lines show the location of prominent moraines including those mapped by Graf (1971) in the Rock Creek drainage. Inset map shows the location of Montana (MT) and Wyoming (WY) highlighted in blue. Elevation data from the U.S. Geological Survey 3D Elevation Program (U.S. Geological Survey, 20171130, USGS 13 arc-second n46w110 1 x 1 degree: U.S. Geological Survey).

terraces are common within these valleys and provide abundant geologic evidence of past glacial activity from the LGM and deglaciation. Higher in the valleys, proximal to cirque headwalls, additional glacial evidence is preserved as sharp-crested moraines, rock glaciers, and protalus lobes (Davis, 1988; Graf, 1971). Most features are mapped within 1-2 km from cirque headwalls and may represent multiple phases of glaciation (Davis, 1988). Based on geomorphologic characteristics, Graf (1971) identified such features in the Beartooths as two periods of late-Holocene



glaciation. However, geochronologic data are limited by relative-age methods and their associated uncertainties (Davis, 1988).

The Rock Creek drainage, a ~25 km long glacial valley, exits from the northeast corner of the Beartooth Range with associated cirque valleys, including Glacier Lake Valley (study area), located along the border between MT and WY and just north of Yellowstone National Park (Fig. 1 & 2). The lower portion of this drainage (herein referred to as the Lower Valley) is fed by five higher-elevation glacial cirques, trends southwest-northeast, and drains into the town of Red Lodge, MT. Sets of lateral moraines extend north from the valley mouth and bound the town on the eastern and western sides. The lateral moraines define an elongate and relatively narrow former glacier, and they are similar in morphology and size to moraines in valleys 25 km to the west beneath Black Butte. Outboard of the moraines, there are fluvially-modified hills unaffected by glacial activity, while outwash sediments between the lateral moraines suggest extensive meltwater drainage from the Lower Valley. A terminal moraine for this glacier is not preserved, likely due to erosion from meltwater flowing north to the Yellowstone River (Fig. 1). Graf (1971) mapped five moraines within the Lower Valley, although preservation is patchy, including stream breaches ranging from 70 m wide to as wide as the valley floor in some cases. Steep sloped, rocky walls define the valley width, which averages 1.5 to 2.0 km at its widest and narrows further upvalley. Occasional large (>1 m) boulders along the valley floor exhibit glacially-faceted surfaces and striations. The Lower Valley is gently sloped, gaining only 800 m of elevation from the Great Plains over 25 km.

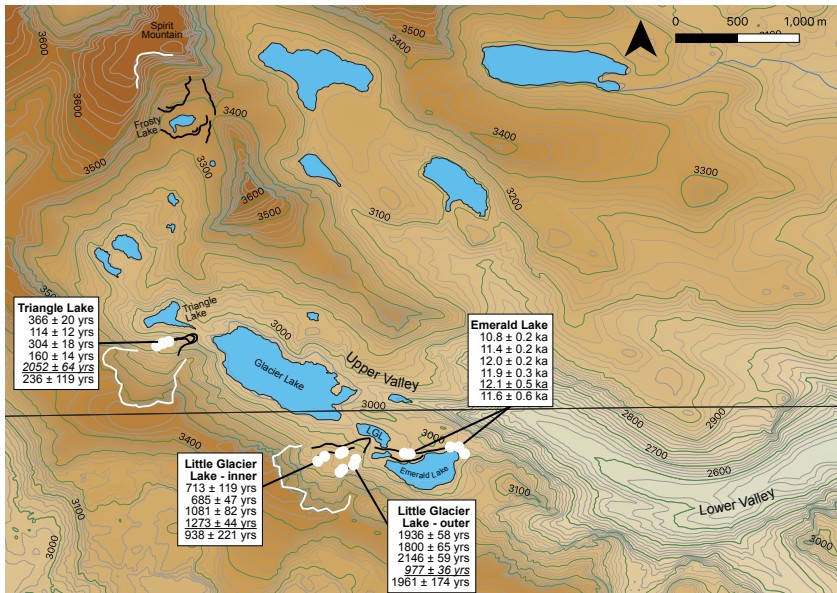

**Figure 2 - Glacier Lake Valley map -** Map of the study area highlighting the names of important locations discussed in the paper. White lines indicate the cirque headwalls. Solid black lines show the location of moraines. Relative elevation shown using the same color scheme as Figure 1. White circles indicate the locations of surface exposure samples with their ages contained in the respective boxes. Mean age and standard deviation for each population located beneath the line in each box. Statistically identified outliers are italicized. LGL =





**Little Glacier Lake. Elevation data from the U.S. Geological Survey 3D Elevation Program (U.S. Geological Survey, 20171130, USGS 13 arc-second n46w110 1 x 1 degree: U.S. Geological Survey).**

There is a steep, 400 m-high transition from the Lower Valley to Glacier Lake valley (herein referred to as the Upper Valley), capped by an exposed, glacially-abraded, bedrock lip. Four lakes are located between 2960-2970

m asl within the Upper Valley (from east to west): Emerald Lake, Little Glacier Lake, Glacier Lake, and Triangle Lake (Fig. 2). Two of the lakes, Emerald Lake and Little Glacier Lake, are bounded by the bedrock lip to the north. To the south, Emerald Lake is bounded by an exposed bedrock cliff while Little Glacier Lake by a bouldery moraine. A deflated and curved boulder-covered moraine exists in the gentle topography between the two lakes (referred to as the Emerald Lake moraine) and is the stratigraphically oldest moraine in the Upper Valley (Graf, 1971). The location

and size of Glacier Lake is controlled by steep rocky walls to the north and south with colluvial fans along the base of the southern wall. Two bedrock knobs separate Glacier Lake from Triangle Lake to the west, with morainal deposits restricting water flow between the two (Graf, 1971). To the northwest of Triangle Lake, a rounded and smoothed bedrock exposure increases elevation as the valley changes orientation to north-south and gains another ~450 m of elevation. A small cirque below the peak of Spirit Mountain (3744 m asl) contains a lake, Frosty Lake, and two

moraines along both the down- and upvalley shores. The downvalley moraine is low-relief, bouldery, and rests on the cirque lip. The upvalley moraine is 50 m tall, sharp-crested, within 25 m of modern ice, and morphologically similar to the moraine at Triangle Lake. No prominent moraines are present between Triangle Lake and those at Frosty Lake.

The cirque near Little Glacier Lake preserves two moraines with an additional moraine in the cirque near Triangle Lake (Fig. 2). The moraines around Little Glacier Lake have high-angle lateral slopes of large (>1 m) angular

boulders mixed with coarse to fine sands and all grain sizes in between (Fig. 3). The toe of the moraine is lower-relief (15 m), dominated by boulders, and rests along the southern shore of Little Glacier Lake. The left-lateral moraine rests on a bedrock outcrop at 3022 m asl (Fig. 3B). Ribbed crests of boulder deposits are found inboard of the left lateral moraine and maintain a similar elevation throughout. Between the lateral moraines, the surface elevation remains fairly consistent, with drastic changes in slope only along the distal sides of the moraines. While the moraine crests

are easily defined, abundant boulder debris coverage within the moraine limits is suggestive of rock glacier activity or rockfall from over-steepened cliff faces. Small patches of modern ice are visible near the headwall but are covered in debris further down slope.

A steep-sloped moraine is found along the southern shore of Triangle Lake, ~1.2 km north of a cirque headwall containing a small (~0.1 km²), debris-covered glacier. The moraine increases in relief from 35 m near the

terminus to 80 m along the lateral aspects and contains abundant coarse sands and gravels among numerous angular boulders (>0.4 m in height). The peak of the moraine is narrow and exhibits minimal deflation, yet occasional patches of grass and soil suggest the surficial deposits are not recent. From the terminus, the moraine curves to the east to intersect a near-vertical cliff face and is covered with boulders, some up to ~10 m in size. The western aspect of the moraine gains 130 m elevation before encountering a glacially-smoothed bedrock knob. Above the bedrock knob, the

moraine continues another 60 m before reaching the glacier.

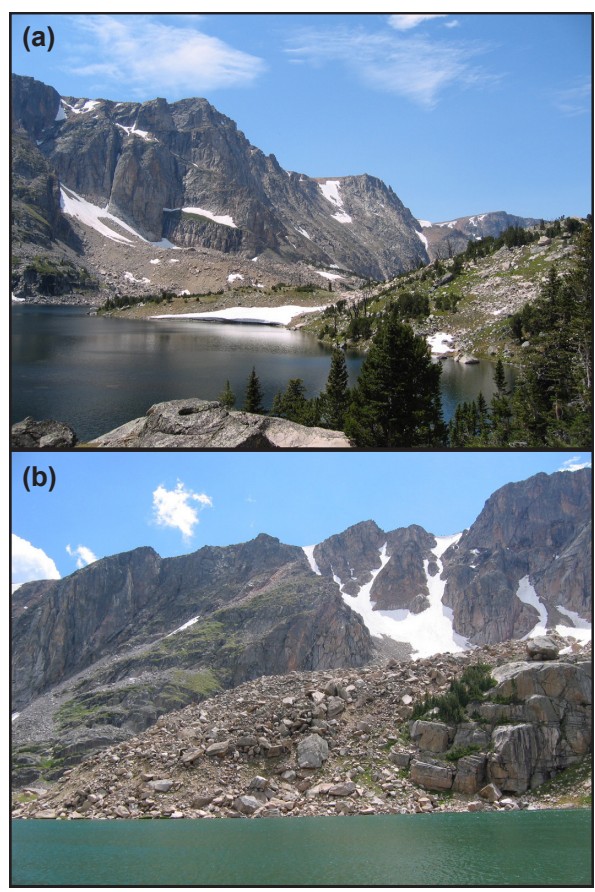

**Figure 3 - Beartooth Moraines – (a) Emerald Lake moraine on the left with the Little Glacier Lake moraines in the distance. (b) Toe and left lateral aspect of the Little Glacier Lake moraine and bedrock knob.**


### 3 Methods

To determine the timing of glaciation within Upper Valley, we sampled glacially-deposited boulders for cosmogenic $^{10}$Be surface exposure dating from four moraines: one at Triangle Lake, two at Little Glacier Lake, and one at Emerald Lake moraine (Fig. 4). Samples were collected over two field seasons in 2006 (Emerald Lake and

Little Glacier Lake) and 2017 (Triangle Lake and Little Glacier Lake) using hammer and tungsten carbide-tipped chisels. Strict sample selection criteria were followed to maximize the accuracy of nuclide accumulation representative of initial deposition by the glacier (Dunai, 2010; Gosse and Phillips, 2001). Boulders selected for sampling exhibited no signs of surficial erosion or pitting that would remove accumulated nuclides. Samples were limited to within the top five centimeters from flat-topped boulders to ensure highest rates of *in-situ* nuclide production. All samples were

collected on or near the mapped moraine crest to minimize potential post-depositional movement and associated reduction of nuclide accumulation on the sampled surface. Each boulder was a minimum of 0.4 m in height with an average boulder height of 1.5 m. Potential boulder burial or exhumation is unlikely given boulder height and morphological characteristics of the moraines. Both the Little Glacier Lake and Triangle Lake moraines exhibit little





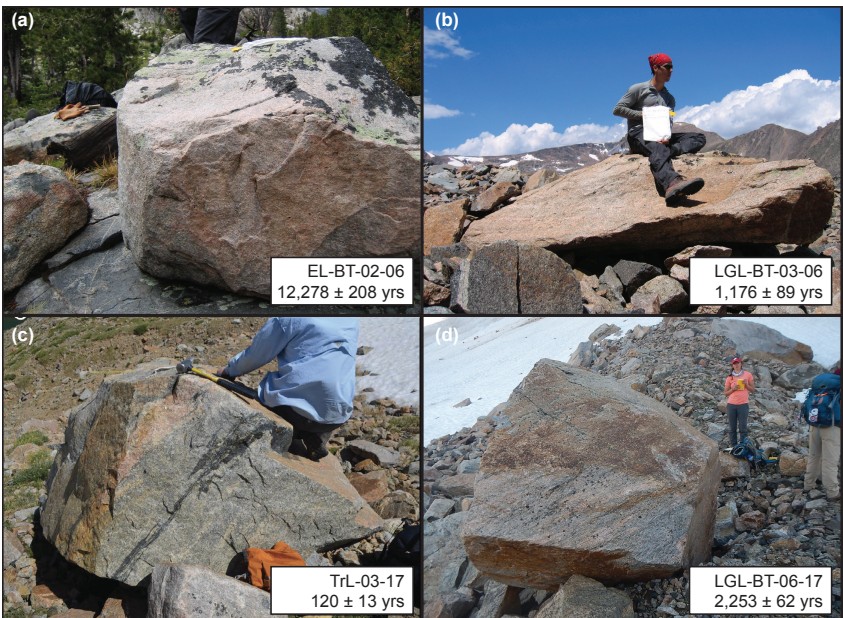

**Figure 4 - Sample boulders - Boulders sampled for [10]Be surface exposure dating discussed in this project. (a) EL = Emerald Lake. (b,d) LGL = Little Glacier Lake. (c) TrL = Triangle Lake. 1-sigma analytic uncertainty shown for each age.**

to no soil cover, which suggests exhumation of boulders from previous cover is unlikely. Latitude, longitude, and elevation were recorded for each sample using a handheld GPS unit and verified using high-resolution DEMs and topographic maps. Surface inclination of each sample was measured using a Brunton compass and topographic shielding was recorded in the field using a clinometer. Shielding scaling factors for each sample were calculated using the CRONUS online Topographic Shielding Calculator (http://stoneage.ice-d.org/math/skyline/skyline_in.html; Balco et al., 2008)

A total of 33 samples were collected over the two field seasons including two samples from the exposed bedrock at Triangle Lake, which exhibited glacial polishing. Eighteen samples were processed for $^{10}$Be extraction that best represent our selection criteria: five boulders from Emerald Lake, four from the inner Little Glacier Lake moraine, four from the outer Little Glacier Lake moraine, and five from Triangle Lake.

Boulder samples were processed for $^{10}$Be extraction at the University of Wisconsin – Madison. The sample processing procedures are adapted from laboratories at Oregon State University, the University of Vermont Cosmogenic Nuclide Laboratory, and the University of New Hampshire (Corbett et al., 2016; Kohl and Nishiizumi, 1992; Licciardi, 2000; Marcott, 2011). Each rock sample was sawed to isolate the top 2-5 cm, crushed and sieved to a 250-710 μm size fraction, magnetically separated, and etched in concentrated HCl and dilute HF/HNO₃ acid solutions. Chemical frothing was performed on all non-magnetic grains in order to isolate quartz from feldspar. The remaining quartz grains were isolated using additional HF/HNO₃ etches. Quartz purity was measured by inductively



coupled plasma optical emission spectroscopy (ICP-OES) at the University of Colorado-Boulder Department of Geological Sciences and the University of Wisconsin-Madison Water Science and Engineering Laboratory.

The chemically etched and pure quartz was dissolved in concentrated HF, after an addition of $^9$Be carrier solution prepared from raw beryl (OSU White standard; $^9$Be concentration of $251.6 \pm 0.9$ ppm) (Marcott, 2011). We

used anion and cation exchange chromatography to separate Fe, Ti, and Al from Be, and BeOH was precipitated in a pH 8 solution. BeOH gels were converted to BeO by heating to 900-1000ºC with a rapid incinerator, then mixed with Nb powder in quartz crucibles, and packed into stainless steel cathodes for accelerator mass spectrometry (AMS) analysis.

All $^{10}$Be/$^9$Be ratios were measured at the Purdue University Rare Isotope Measurement Laboratory (PRIME

Lab) and normalized to standard 07KNSTD3110, which has an assumed $^{10}$Be/$^9$Be ratio of $2.85 \times 10^{12}$ (Nishiizumi et al., 2007). Sample ratios were corrected using batch-specific blank values that ranged from $2.39 \times 10^{-15}$ to $4.19 \times 10^{-15}$ (n = 3; Table 1). $^{10}$Be concentrations presented in Table 1 are corrected from process blanks.

Exposure ages are calculated using version 3.0 of the original online calculator as described in Balco et al. (2008; herein referred to as Version 3.0) and the time-dependent scaling scheme of Lifton et al. (2014; LDSn). We

use the $^{10}$Be production rate in quartz as determined from Promontory Point, UT (PPT; Lifton et al., 2015) based on proximity to the study site (~700 km) and high confidence in the quality of constraining samples. We note that a recent review of Western U.S. surface exposure ages follows similar methods for recalculating previous published data (Laabs et al., 2020; Licciardi and Pierce, 2018) including those for the GYGS. Wide-ranging rates of surface erosion experienced by glacial erratics are suggested in the literature with rates within the order of 0.0 to 0.1 cm ka$^{-1}$

(Ballantyne and Stone, 2012). Based on the presumed young age of our sampled deposits, we assume surface erosion of the boulders was negligible and therefore do not account for it in our calculations. We follow the procedures outlined in Laabs et al. (2020) and assume the effects of snow cover to be negligible. Individual exposure ages are reported as years before collection date (per Version 3.0 calculations) and with 1-sigma analytical uncertainty. Moraine ages are reported as the arithmetic mean and standard deviation of the boulder ages to present a conservative uncertainty

estimate. We note that distribution ages reported using error-weighted mean and uncertainty tend to favor younger exposure ages with lower analytical uncertainty which are more likely to be influenced by geological processes (e.g., surface erosion, post-depositional movement) leading to erroneously young ages (Barth et al., 2019; Laabs et al., 2020). Outliers are identified using Chauvenet's criterion and interquartile range tests.

**4 Results**

Resulting $^{10}$Be exposure ages demonstrate glaciation within the Upper Valley from the latest Pleistocene and the late Holocene with all ages in stratigraphic order. Five exposure ages from the stratigraphically oldest moraine at Emerald Lake range from $11.3 \pm 0.2$ to $13.0 \pm 0.5$ ka (Table 1). Together these samples have a mean age and standard deviation of $12.5 \pm 0.7$ ka. We note that sample EL-BT-01-06 ($11.3 \pm 0.2$ ka) is 0.9 ka younger than the next youngest

sample (EL-BT-02-06; $11.4 \pm 0.2$ ka), yet cannot be rejected using either of the statistical tests. Removing sample EL-BT-01-06 from the population increases the mean age to 12.8 ka which falls within 1-sigma uncertainty of our reported age and therefore does not change our interpretations. Age of the Emerald Lake moraine calculated using the Titcomb



Lakes, WY production rate (TL; Gosse et al., 1995) yields a mean age and standard deviation of 11.6 ± 0.6 ka. This age difference of 0.9 ka between production rates has implications for the paleoclimatic interpretation. Here we discuss the paleoclimatic implications of both ages as it relates to glaciation within the Beartooth Mountains.

Samples (n = 4) from the Little Glacier Lake outer moraine yield exposure ages ranging from 1,061 ± 39 yrs to 2,253 ± 62 yrs (Table 1). Sample LGL-BT-07-17 (1,061 ± 39 yrs) is identified as an outlier and excluded from the population. The mean age and uncertainty of the remaining distribution is 2,063 ± 178 yrs.

The inner Little Glacier Lake moraine yielded four exposure ages ranging from 724 ± 49 yrs to 1,354 ± 47 yrs (Table 1). A ~600 yr range exists between the oldest and youngest age of the population, however, no samples are rejected using the statistical tests described. These samples were likely affected by geologic processes. One possible explanation is some boulders contain accumulated nuclides from previous periods of exposure and are perhaps sourced from rock fall from the nearby cirque headwall and cliff faces. If this is the case, the youngest ages from the moraine are most likely to represent a period of glaciation. Based on geomorphological characteristics, including the bouldery nature and steep toe of the moraine, another possibility is that this moraine is the product of rock glaciation with lower rates of erosion less likely to remove previously accumulated nuclides from past periods of exposure. The sample population as a whole demonstrates a multi-modal distribution, which could be interpreted as numerous periods of glaciation. However, we offer a more conservative interpretation and consider the youngest age as a minimum limiting age of glaciation for this particular glacier in the Upper Valley.

Five samples from the moraine near Triangle Lake yield exposure ages ranging from 120 ± 13 yrs to 2,156 ± 67 yrs. Sample TrL-07-17 (2,156 ± 67 yrs) is identified as an outlier using both previously mentioned statistical tests. The four remaining samples exhibit an approximate bimodal distribution, but overlap within 2-sigma uncertainty. Therefore, we interpret the mean age of 249 ± 126 yrs for all four samples.

Choice of production rate for our ages has little effect on the Little Glacier Lake and Triangle Lake moraines with each moraine age changing by fewer than 100 years (Table 2).

## 5 Discussion

### 5.1 Variable rates of western U.S. glacier retreat from the LGM to the Younger Dryas

During the last deglaciation the timing and rate of retreat of western U.S. glaciers varied spatially (Laabs et al., 2020). Broadly, glacial retreat from LGM margins in the western U.S. began between 22 ka and 18 ka (Shakun et al., 2015; Young et al., 2011). Glaciers from the northern GYGS fall within this window of deglacial onset, with two terminal moraines yielding $^{10}$Be exposure ages of 19.8 ± 0.9 ka and 18.2 ± 1.3 ka (Fig. 5; Licciardi and Pierce, 2018). While no geochronologic control exists for the lateral moraines near Red Lodge, a lack of moraines outboard of them suggest they represent a terminal position of the glacier. They are also morphologically similar to other LGM-mapped moraines within the Beartooths (Black Butte; 25 km west of Red Lodge), and are proximal to the dated moraines at Clarks Fork (19.8 ± 0.9 ka) and Pine Creek (18.2 ± 1.3 ka) that are part of the same regional glacial system. Therefore, for our purposes, we assume the age, and therefore the onset of deglaciation, of the Red Lodge moraines to fall within the time window of 22 ka to 18 ka.





| Table 1 - Sample data and [10]Be concentrations | | | | | | | | |
|---|---|---|---|---|---|---|---|---|
| Sample name | Latitude (DD) | Longitude (DD) | Elevation (m) | Thickness (cm) | Shielding | Date | Conc. (atoms g[-1]) | Conc. unc. (atoms g[-1]) |
| *Triangle Lake[a]* | | | | | | | | |
| TrL-02-17 | 45.00993 | -109.55342 | 3064 | 2 | 0.937 | 2017 | 17996 | 992 |
| TrL-03-17 | 45.01003 | -109.55327 | 3056 | 2 | 0.939 | 2017 | 5229 | 560 |
| TrL-05-17 | 45.01023 | -109.55276 | 3049 | 5 | 0.939 | 2017 | 14573 | 850 |
| TrL-06-17 | 45.01021 | -109.55286 | 3049 | 4 | 0.939 | 2017 | 7370 | 647 |
| TrL-07-17 | 45.01039 | -109.55219 | 3058 | 2 | 0.939 | 2017 | 84217 | 2632 |
| *Little Glacier Lake - outer[a]* | | | | | | | | |
| LGL-BT-02-17 | 44.9983 | -109.533 | 3042 | 2 | 0.981 | 2017 | 81645 | 2443 |
| LGL-BT-04-17 | 44.99907 | -109.53428 | 3023 | 2 | 0.951 | 2017 | 53327 | 1863 |
| LGL-BT-05-17 | 44.9989 | -109.5346 | 3045 | 2 | 0.96 | 2017 | 74396 | 2668 |
| LGL-BT-06-17 | 44.9981 | -109.5369 | 3057 | 2 | 0.909 | 2017 | 85516 | 2366 |
| LGL-BT-07-17 | 44.9986 | -109.5362 | 3060 | 2 | 0.938 | 2017 | 45089 | 1664 |
| *Little Glacier Lake - inner[b]* | | | | | | | | |
| LGL-BT-01-06 | 44.99698 | -109.53462 | 3094 | 2 | 0.891 | 2006 | 33396 | 5584 |
| LGL-BT-02-06 | 44.99728 | -109.53433 | 3075 | 2 | 0.891 | 2006 | 31674 | 2156 |
| LGL-BT-03-06 | 44.99772 | -109.53322 | 3057 | 2 | 0.935 | 2006 | 47683 | 3620 |
| *Emerald Lake[b]* | | | | | | | | |
| EL-BT-01-06 | 44.99887 | -109.52185 | 2999 | 2 | 0.969 | 2006 | 477331 | 8107 |
| EL-BT-02-06 | 44.99957 | -109.52243 | 2993 | 2 | 0.989 | 2006 | 517609 | 8749 |
| EL-BT-03-06 | 44.99957 | -109.5233 | 2990 | 2 | 0.989 | 2006 | 537278 | 9494 |
| EL-BT-06-06 | 44.99892 | -109.52802 | 2989 | 2 | 0.977 | 2006 | 528085 | 14047 |
| EL-BT-07-06 | 44.99885 | -109.52743 | 2991 | 2 | 0.977 | 2006 | 533612 | 21843 |
| *Blanks* | | | | | | | | |
| Blank-004 | - | - | - | - | - | - | 54395 | 10291 |
| Blank-005 | - | - | - | - | - | - | 37559 | 8405 |
| Blank-024 | - | - | - | - | - | - | 30634 | 5901 |
| [a]Samples that used Blank-024 in calculations | | | | | | | | |
| [b]Samples that used Blank-004 and Blank-005 in calculations | | | | | | | | |


Rapid retreat of the Clarks Fork glacier to the Crandall Creek moraine between $19.8 \pm 0.9$ ka and $18.2 \pm 0.8$ ka was followed by the construction of multiple recessional moraines as retreat slowed (Licciardi and Pierce, 2018).

Initial, rapid retreat of the Clarks Fork glacier is hypothesized to have occurred as migration of the Yellowstone Ice Cap to the southwest created a precipitation shadow for the northeast portion of the GYGS thus affecting glacial mass balance negatively (Licciardi and Pierce, 2018, 2008). Synchronous changes in North American precipitation patterns were affecting glacial mass balance in the central north region of the U.S. (Lora and Ibarra, 2019). These changes in precipitation include migration of the dominant atmospheric jet stream (Lora et al., 2017) and changes in the

seasonality of precipitation delivery to certain regions (Lora and Ibarra, 2019). Slower retreat of the Clarks Fork glacier during the Late Pinedale (16-13 ka) is attributed to a reduction of the precipitation shadow in the northeast GYGS as the Yellowstone Ice Cap continued its migration to the southwest (Licciardi and Pierce, 2018). After ~14 ka a more regionally coherent response of glacier retreat occurred in response to warming from increased atmospheric greenhouse gas concentrations (Marcott et al., 2019, 2014; Shakun et al., 2015).

Graf (1971) mapped five moraines within the Lower Valley between our prescribed LGM moraines and the Upper Valley which mark the location of the glacier as it retreated from its terminal position (Fig. 1). Based on geomorphological evidence such as distance regression, soil development, and shape index, the two furthest down valley moraines were associated with Bull Lake glaciation and the three more upvalley moraines as Pinedale terminal moraines. Furthermore, Graf (1971) assigned the Pinedale moraines correlative ages with other Western U.S. deposits

ranging from 23.0 ka for the outermost to 11.5 ka for a moraine found near the steep transition from Lower to Upper Valley. However, based on the reasons described above, we argue all deposits within the Lower Valley are more likely



representative of post-LGM deglaciation similar to proximal glacial valleys within the GYGS (Laabs et al., 2020; Licciardi and Pierce, 2018, 2008). Additionally, the age of glacial retreat we obtained for the Emerald Lake moraine (12.5 ± 0.7 ka) inboard from the Graf (1971) 11.5 ka Lower Valley moraine suggests stabilization of the glacial prior

to 12.5 ± 0.7 ka and is therefore in stratigraphic disagreement with this assigned age. As such, the Lower Valley deposits must be older than 12.5 ± 0.7 ka, and represent a period of glaciation between the end of the LGM (22 ka to 18 ka) and formation of the Emerald Lake moraine (Fig. 5).

The five Lower Valley moraines between Red Lodge and the Upper Valley suggest multiple periods of stagnation or glacial readvance during the last deglaciation. A lack of numerical age control limits our ability to assess

the rate and timing of retreat within the Lower Valley. However, stratigraphic relationships of the moraines along with chronologic control higher in the valley permit a certain amount of bookending.

The Emerald Lake moraine marks the next geochronologically-constrained position of the glacier following the LGM. Using the time window of 22 ka to 18 ka for the onset of LGM deglaciation in Red Lodge and the age of the Emerald Lake moraine as limiting ages we determined a maximum and minimum mean retreat rate of the glacier

of 4.9 m/yr (18.0 – 12.5 ka) and 2.8 m/yr (22.0 – 11.6 ka), respectively. These values are lower than those recorded for the nearby Clarks Fork glacier, which lost 75-90% of its LGM length between 19.8 ± 0.9 ka and 18.2 ± 0.8 ka at a mean retreat rate of 28.0 m/yr (Licciardi and Pierce, 2018).

We hypothesize that retreat of the Rock Creek glacier from its LGM limit began later in the window of LGM retreat (22 to 18 ka) similar to the Pine Creek moraine (18.2 ± 1.3 ka), experienced slower retreat similar to that of the

Clarks Fork glacier after abandoning the Crandall Creek moraine (18.2 ± 0.8 ka), thus allowing for the formation of morainal features within the Lower Valley, and eventually left the Lower Valley in response to rising atmospheric greenhouse gases.

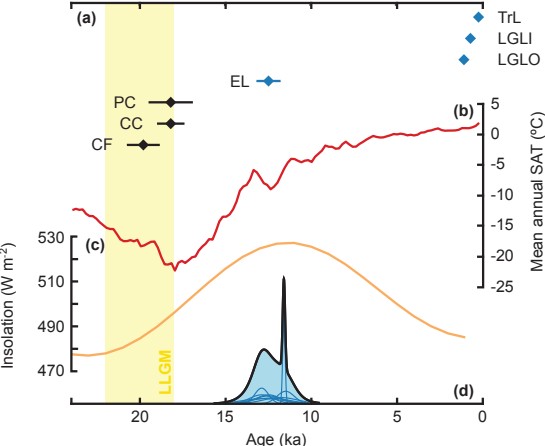

**Fig. 5 - LGM, Deglacial, and Holocene- Diamonds indicate moraine ages. (a) Blue diamonds are ages from this study. TrL = Triangle Lake, LGLI = Little Glacier Lake inner moraine, LGLO = Little Glacier Lake outer moraine. Black diamonds are ages from Licciardi & Pierce (2018). PC = Pine Creek moraine, CC = Crandall Creek, CF = Clarks Fork. (b) Red line is mean annual surface temperature for our study area taken from**





Osman et al. (2021). **(c) Orange line is mean annual insolation for 45º N. (d) Bottom probability distribution shaded in blue is for the compilation of western U.S. Younger Dryas moraines taken from Marcott et al. (2019), Licciardi and Pierce (2018), and the Emerald Lake moraine from this study. Yellow box highlights the timing of the local LGM from 22 ka to 18 ka.**

### 5.2 The Younger Dryas

Presence of the Emerald Lake moraine shows the location of the glacier in the latest Pleistocene, older than the previously proposed Neoglacial age (Graf, 1971), and marks a pinning point of glaciation during retreat from the LGM limits. Additionally, these data contribute to a growing data set of western U.S. moraine ages supporting a regional response of mountain glaciers to the YD (Marcott et al., 2019),

The Emerald Lake moraine marks the first location of the glacier margin in the Upper Valley ~27 km upvalley from the LGM position. Presence of the moraine indicates stillstand or readvance of the glacier during the latest Pleistocene, perhaps in response to Younger Dryas (YD) cooling (12.9 – 11.7 ka; Davis et al., 2009; Gosse et al., 1995). Geochronologic studies from glaciers in additional western U.S. mountain ranges identify similar glacier responses to YD cooling (Gosse et al., 1995; Licciardi and Pierce, 2008; Marcott et al., 2019) while other paleoclimate proxies identify a YD signal in the region (Mix et al., 1999; Praetorius et al., 2020, 2015; Vacco et al., 2005). However, uncertainty of the exposure ages, and differences in production rate calculations, prevent conclusively attributing formation of the Emerald Lake moraine to the early- or later-portions of the YD stadial. Two potential scenarios arise: (1) deposition of the moraine early in the YD cooling with glacier retreat continuing throughout the stadial, and (2) deposition in response to YD cooling and retreat following abrupt warming at the end of the YD.

The age of the Emerald Lake moraine reported using the PPT production rate places retreat of the glacier from this location 12.5 ± 0.7 ka with glacial stagnation occurring prior to this time (Fig. 5). Morainal deposits near the Lake Solitude cirque lip in the nearby Teton Range (~175 km to the south) are dated to 12.9 ± 0.7 ka (Licciardi and Pierce, 2008; recalculated in Licciardi and Pierce, 2018). Together the ages of the Emerald Lake and Lake Solitude moraines imply a glacier response to the YD cooling with retreat during, or after, the stadial. A compilation of YD-age deposits from the western U.S. (Marcott et al., 2019), calculated using the PPT production rate and LSDn scaling scheme, highlight 9 separate moraine exposure ages (including the Emerald Lake and Lake Solitude moraines) between 12.9 ± 0.7 ka (Licciardi and Pierce, 2008) and 11.5 ± 0.5 ka (Marcott et al., 2019). Mean age and standard deviation of these moraines are 12.3 ± 0.6 ka, while probability density indicates a higher peak earlier in the stadial (12.8 ka). All moraines in the compilation exist between 37-45º N and above 2765 m elevation. Taken together, these exposure ages suggest a regional glacial response to the YD stadial, with variability in timing of morainal abandonment– potentially driven by differences in precipitation patterns (Lora and Ibarra, 2019), atmospheric lapse rate (Loomis et al., 2017; Zhang et al., 2019), or topographic control (Barr and Spagnolo, 2015).

Age of the Emerald Lake moraine calculated using the TL production rate (11.6 ± 0.6 ka) indicates retreat of the glacier later in the YD. Even with consideration of age uncertainty, the reported exposure age highlights moraine abandonment ~700 years after the onset of the YD stadial at a minimum (Fig. 5). As such, the glacier potentially stabilized in response to abruptly suppressed temperatures, maintained its position for > 700 years, and retreated as temperatures recovered at the end of the stadial. However, the deflated nature of the Emerald Lake moraine lies in contrast to the much younger Triangle and Frosty Lake moraines, as well as to the older LGM moraines found at

Clarks Fork and Black Butte. Similar low-profile morainal deposits associated with mountain glaciation are attributed to high-frequency glacier length variability during deglaciation (Barth et al., 2018). Therefore, the subdued profile of

the Emerald Lake moraine suggests any period of stagnation was short-lived and/or characterized by low depositional rates. Any forcing of this response had minor impact in the overall trend of glacier retreat.

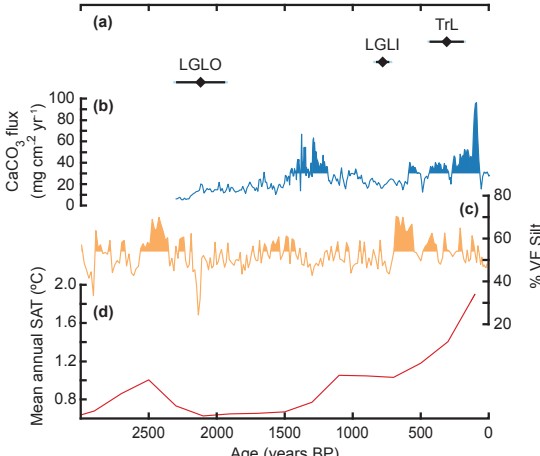

**Fig. 6 - Neoglacial – (a) Ages for the late-Holocene moraines from this study adjusted to reflect years before**
**present. Black lines indicate 1-sigma analytic uncertainty. Light blue lines in the ages include the production**
**rate uncertainty added in quadrature. (b) Blue line shows the CaCO₃ flux from Harrison Lake in Glacier**
**National Park (Munroe et al., 2012). Shaded areas indicate greater than mean values for the time series and**
**interpreted as increased glacial activity in the original study. (c) Orange line shows the percent very fine silt**
**fraction from Cracker Lake in Glacier National Park (Munroe et al., 2012). Shaded areas indicate greater than**
**mean values and increased glacial activity. (d) Red line is the mean annual surface temperature for our study**
**region taken from Osman et al. (2021).**

### 5.3 Constraints on Neoglaciation and the Little Ice Age

Ages from the Little Glacier Lake and Triangle Lake moraines indicate Upper Valley glacier activity within the
late Holocene. Glaciation in the western U.S. is suggested to have reached a minimum in the early Holocene due to high Northern Hemisphere summer insolation (Fig., 6; Porter and Denton, 1967). Multiple archives across the northern hemisphere highlight a readvance of glaciers in the later Holocene (Munroe et al., 2012; Porter and Denton, 1967; Reyes et al., 2006; Solomina et al., 2015) due to declining summer temperatures (Marcott et al., 2013; McKay et al., 2018). Late-Holocene, Neoglacial deposits are found in numerous mountain ranges across the western U.S. and
Canada, Greenland, Europe, and Asia (Briner and Porter, 2018; Graf, 1971; Marcott et al., 2019; Munroe et al., 2012; Reyes et al., 2006; Solomina et al., 2015).

Age of the Little Glacier Lake outer moraine indicates conditions within the Upper Valley had become favorable for glacier activity by 2,063 ± 178 years ago with glaciation likely occurring prior to this time. This age also implies that subsequent glaciation (e.g., LIA) did not exceed this boundary which therefore represents the maximum late-
Holocene glacial extent. Based on geomorphic evidence and variability among exposure ages, we suggest that the Little Glacier Lake inner moraine represents rock glaciation which form through transitional glacial and periglacial



processes (Petersen et al., 2020). Conservatively, we assume the youngest ages within the distribution act as minimum-limiting constraints on the timing of glaciation after the abandonment of the outer moraine. Therefore, continued and variable glacial-periglacial conditions for the Little Glacier Lake rock glacier is demonstrated by the youngest
exposure ages from the inner deposit (724 ± 49 yrs; Fig. 6).

In nearby Glacier National Park, MT (~500 km north of the Beartooth Mountains) proglacial lake sediments are interpreted to indicate glacier fluctuations throughout the Holocene including phases of glacier advance 2,300 and 1,500 cal year BP and again after 700 cal year BP (Fig. 5; Munroe et al., 2012). Paleo-proxy reconstructions of temperature and precipitation for the region indicate lower than average temperatures and increased winter
precipitation ~1,600-1,200 years ago with continued decreased temperatures ~700 years ago (Trouet et al., 2013; Viau et al., 2012). Such conditions present favorable climates for glaciation and correlate with the timing of Little Glacier Lake activity.

Continuation of climate/glacier variability recorded in ages of the Little Glacier Lake deposits and regional paleo-proxies culminated with the glaciation recorded in the Triangle Lake moraine. Morainal abandonment of the Triangle
Lake glacier 249 ± 126 yrs ago marks the end of substantial moraine construction within the Upper Valley. Similarities between the Triangle Lake and the upvalley Frosty Lake moraine morphologies imply similar age and behavior of the two glaciers. Both moraines are large (>50 m relief) likely from either long duration of moraine construction at the boundary or high depositional rates. We suggest late-Holocene reactivation of the Triangle Lake glacier was synchronous with the onset of glaciation recorded in the Little Glacier Lake deposits as both responded to the same
forcing. Unlike at Little Glacier Lake, the Triangle Lake glacier maintained its position and positive mass balance longer, thus constructing a more substantial moraine and staving off rock glaciation, which can occur as a periglacial feature (Giardino and Vitek, 1988; Knight et al., 2019).

A recent study of glaciers in New Zealand highlight the inverse hemispheric relationship as those glaciers reached maximum lengths early in the Holocene and retreated ever since (Dowling et al., 2021). It is likely that glaciers and
periglacial features in the higher elevations of the Beartooth Mountains reactivated in response to late-Holocene low NH insolation levels, and provide evidence of cooler, and more favorable, climates in the last half of the Holocene. Though proximal, differences in Little Glacier Lake and Triangle Lake glacier response to climate forcing could be driven by the influence of higher, and more shaded, headwalls surrounding the Triangle Lake glacier. Similar influence of topography have been shown to influence alpine glacier advances (Barr and Spagnolo, 2015). Taken together, the
Little Glacier Lake and Triangle Lake exposure ages suggest reactivation of glaciers within the Upper Valley from their early Holocene minima during the Neoglaciation prior to 2,063 ± 178 yrs with deglaciation occurring as late as 249 ± 126 yrs. Exposure ages from the Little Glacier Lake inner deposit support regional studies suggestive of glacier fluctuations and climate variability throughout the Neoglaciation.

**6 Conclusion**

New [10]Be surface exposure ages from the Rock Creek drainage in the eastern Beartooth mountain range yield new insight into the timing of western U.S. glaciation. After likely retreating from the LGM limit between 22 ka and 18 ka, the glacier stabilized in the Upper Valley early in the Younger Dryas stadial as evidenced by the Emerald Lake



moraine (12.5 ± 0.7 ka). This age redefines the age of the deposit, which was initially suggested to be Neoglacial in
age by Graf (1971), and fits a trend of previously assigned Neoglacial deposits being geochronologically constrained
to latest Pleistocene and earliest Holocene glaciation (Marcott et al., 2019). Evidence of Neoglaciation is found within
the Upper Valley in the presence Little Glacier Lake outer moraine that yields an age of 2,063 ± 178 yrs, and continued
glacial-periglacial rock glacier fluctuations occurring throughout the late Holocene (724 ± 49 yrs). These deposits
support evidence that climatic conditions within the western U.S. were favorable for glaciation at high elevations in
the late Holocene (Marcott et al., 2019; Solomina et al., 2016). Glaciation continued within the Upper Valley until
retreat of the Triangle Lake glacier 249 ± 126 yrs, with only ephemeral glaciers and ice patches remaining today.

## Data availability

Data from this study including those used in the figures are available in the Supplement.


## Supplement link

The supplement related to this article is available online.

## Author contributions

AMB: Conceptualization, Investigation, Writing – Original Draft, Visualization, Funding acquisition. EGC:
Investigation, Writing – Review and Editing. CV: Investigation, Writing – Review and Editing. SAM:
Conceptualization, Investigation, Writing – Review and Editing, Funding acquisition. JDS: Conceptualization,
Investigation, Writing – Review and Editing. MWC: Investigation, Writing – Review and Editing.

## Competing interests


The authors declare that they have no conflict of interest.

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
