# Peer review of "10Be age control of glaciation in the Beartooth Mountains, USA from the latest Pleistocene through the Holocene"

_Geochronology, 2022_

## Author Comment (AC1)

Barth et al. (2022)
Geochronology Review
Anonymous Referee #1

This paper presents new cosmogenic 10Be exposure ages from moraines in the Glacier Lake Valley in the Beartooth Mountains of Montana/Wyoming. The ages span from the Younger Dryas (as interpreted) to very recent, and will provide a useful new chronology for moraines in this location.

However, I do have concerns about the paper in its present form. The manuscript feels like a draft and it requires some significant work before it can be acceptable for publication. First, there are inconsistencies in the way ages are calculated, reported, and interpreted throughout the paper. I have elaborated on these issues below, but they make the paper confusing and difficult to follow at present. Second, I am concerned that the precision of the ages is over-interpreted, and specifically that the attempt to place the Emerald Lake moraine within the Younger Dryas with centennial-scale precision is not warranted. Third, the presentation of the paper needs improvement, including both the writing and the figures. I am confident that all of these issues can be addressed and the manuscript can become acceptable for publication, but the changes required are Major because they touch on all parts of the paper and the underlying age calculations.

My primary concern is that the exposure ages underpinning the study are unclear. The paper seems to be mixing production rates (that are not explained in the Methods), and discussing ages in the text that contradict those in the figures. In addition, there is some confusion about how clusters of boulder ages are being interpreted, and I am concerned that different approaches are taken for different clusters.

The text in the Results describes ages that do not match Fig. 1 (and are not reported in Table 1). For example, at L218 the ages for the Emerald Lake moraine are reported as between 11.3 and 13.0 ka, but on Fig 1 are reported as between 10.8 and 12.1 ka. This is a recurring source of confusion throughout the paper, affecting all the ages and interpretations. The text around L223-224 indicates that different production rates are being used – why is this not explained in the Methods? Mixing production rates is confusing and inappropriate. My understanding is that Fig. 1 is using an old production rate from 1995, as indicated at L223 (because the mean age of 11.6 matches Fig. 1). If this is correct, then a better approach would be to select the most appropriate production rate, use it consistently so the text and figures match, and somewhere within the paper address how the uncertainty on the production rate propagates into uncertainty on the exposure ages.

I'm sure this confusion can be resolved by calculating the ages consistently throughout the paper. However, this does lead me to some further concerns about how the clusters of ages are interpreted.

*Thank you for acknowledging this error in this initial submission. We believe the reviewer is referring to the ages in Figure 2 as Figure 1 does not report any ages. The ages reported in Fig. 2 were mistakenly submitted using the Titcomb Lake (TL) production rate from Gosse et al. (1995), and were meant to report ages using the Promontory Point (PPT) production rate primarily discussed in the manuscript. This error is now corrected with Fig. 2 ages calculated with PPT.*

*Further confusion stems from the discussion in section 5.2 where age of the Emerald Lake is discussed using both the PPT and TL production rate. While our overall method was using PPT (as discussed in our Methods section) we thought it more comprehensive to acknowledge different production-rate specific interpretations of the data. However, we now accept this approach merely created confusion and not our stated goals. As such, we chose to remove all reference to the TL production rate and age of the Emerald*

*Lake moraine using it. We believe this will clarify much of the confusion, and issues, described in this review.*

I am concerned that section 5.2 overstates what can be interpreted from the ages given their uncertainties. It's also not clear what the purpose of this section is. If it's to suggest that the Emerald Lake moraine likely coincides with the Younger Dryas, then I agree but this point probably does not require a section. If it's to interpret where the ages fall within the Younger Dryas, then I do not think this is possible given the uncertainties on the dating. Again, the ages need to be sorted out. At L326-328 two scenarios are offered: deposition at the start or throughout the YD, but the mean age given in Fig. 1 is 11.6 ka, i.e., just after the YD entirely, which seems contradictory.

*Any ages calculated using the TL production rate are now removed therefore reducing any confusion in that regard. The intention of section 5.2 is to discuss the correlation between the age of the Emerald Lake moraine with the timing of the Younger Dryas stadial and potential forcings. Additionally, we intend to describe the Emerald Lake moraine in context of similarly-aged moraines from the region. The larger compilation of "YD-aged" moraines from the western United States indicates a larger, regional glacier response to the abrupt climate change event, while hypothesizing a non-uniform retreat of those glaciers due to other controls (e.g., topography, precipitation, etc.). With removal of discussion involving Emerald Lake ages calculated using the TL production rate more emphasis is given to this aspect of section 5.2 and simplifies the argument.*

The next paragraph opens giving the average age as 12.5 ka, and this magnitude of discrepancy undermines any attempt to place the moraine inside the YD interval at a sub-millennial resolution.

*We believe the discrepancy the reviewer is referring to is the mismatch between the reported 12.5 ka age in the text with that of Fig. 2 where the error is now corrected and mismatch resolved.*

The age given in Fig. 1 then appears at L342 and the data are reinterpreted again, which feels like reading two different papers in parallel.

*Any discussion referencing TL and interpretations of the data are now removed to limit confusion.*

Either way, I don't think the age precision warrants an interpretation of when the moraine formed within the YD. This claim comes up again in the Conclusion at L413 ("early in the Younger Dryas").

*Agreed. The line is now changed to: "...the glacier stabilized in the Upper Valley during the Younger Dryas stadial...".*

Along the same lines, I am sceptical about the claim that exposure ages show regional variability in timings within the Younger Dryas (paragraph ending at L341). If the uncertainties on the ages calculated using different production rates are as large as the duration of the YD itself, I am not convinced that we can resolve much finer-resolution differences in timing across the region.

*All ages in the manuscript are calculated using PPT and reference to multiple production rates removed to reduce confusion. We maintain the possibility of regional variability in response to the YD stadial as evidenced by the range of ages described in the compilation (12.9±0.7 ka to 11.5±0.5 ka) and choose to present it as a hypothesis. However, we also acknowledge uncertainties within the ages prevent us from conclusively defining such variability with the data. Therefore, we added the line to qualify this point: "However, multi-centennial scale uncertainties within each age prevent conclusive attribution of moraines to early or late periods of the YD."*

Aside from the production rates, I am concerned that something does not add up in the way the clusters of boulder exposure ages are being interpreted. Figure 6 is the issue here – the ages for LGLI given in the text are a little older than those shown in Fig. 1, yet the point for LGLI in Fig. 6 is placed younger than the mean age given for LGLI. I understand that different production rates are being mixed, but I think there is an additional difference in the way LGLI and LGLO are being handled. Please clarify. Is Fig. 6 plotting the mean age for LGLO but the youngest age for LGLI? It's not clear what is happening here because L229-231 states that none of the LGLI ages are rejected as outliers, but on Fig. 6 a very precise and comparatively young age is used instead. Perhaps I am missing something, but if so then the text needs to be clearer.

*The ages for LGLI in Fig. 2 are incorrect using the TL production rate – that is now corrected. Additionally, the mean ages in Fig. 2 are now removed per Reviewer #2's comments. Due to variability within the late-Holocene deposits, moraine ages are now discussed using the range of ages opposed to means.*

Subsequently, the text claims that the young moraine ages correlate with the lacustrine records presented in Fig. 6 (L386-387), but it doesn't look like it to me. The LGLI age plotted seems to fall in between the two main shaded peaks in the blue CaCO3 flux curve. Which itself seems to be somewhat anti-correlated with the orange %VF silt curve?

*The lacustrine data from Munroe et al. (2012) presented in Fig. 6 are interpreted to highlight periods of glacial advances as indicated by the shaded regions under the curve. As exposure ages from moraines are interpreted to reflect the onset of moraine abandonment (i.e., deglaciation) we expect that the ages would fall in between the shaded areas of lacustrine data. Furthermore, the lag between glacial advance peaks between the CaCO$_3$ flux data and the %VF silt data contribute to our earlier point regarding regional, small-scale variability between glacier response to climate forcing. We will expand upon this point in the revised text to reduce confusion and clarify support for the regional variability argument.*

Those are my major concerns, and my recommendation is that they need to be addressed before the paper can be acceptable. Below are more minor comments that I hope will help to improve the paper's clarity and impact.

Some of the referencing could give credit to earlier, classic studies. For example around L212; work was done before Barth et al. (2019) to suggest that younger exposure ages from moraines are likely to be biased by incomplete-exposure effects (e.g., erosion, exhumation, toppling, etc). Some original citations could be acknowledged here. Also at L268-270; important work was done before that of Lora and Ibarra, attributing past changes in North American hydroclimate to migration of the jet stream. To my understanding this hypothesis was presented as early as the 1980s. Please also ensure that the referencing is done properly, e.g., I can't find Osman et al. (2021) in the reference list, but it is cited in the figure captions.

*Additional appropriate citations are now added.*

The Methods could be written more precisely:

- L183: How concentrated/dilute were the acids?
  *More specific lab processing information is added.*
- L185: How many rounds of HF/HNO3 etches were performed?
  *Same as above.*

- L200: Specify the reference production rate used – what is the actual rate in atoms/g/yr? Furthermore, I advise against using multiple production rates throughout the paper, but if you're going to do that then please explain it in the Methods.
  *The reference production rate for PPT is added, but no discussion for multiple production rates is necessary as our previous comments discuss.*
- L204: Do Ballantyne and Stone (2012) constrain boulder-surface erosion rates between 0 and 0.1 cm/kyr? I am not sure they do.
  *Ballantyne & Stone (2012) was used as a reference for erosion rates within this range. However, we changed the sentence to reflect sensitivity tests of erosion using a range of erosion rates.*

Much of the writing could be improved, and the paper needs a careful check for readability, language and grammar. Some illustrative examples:

- "Glacial" is used in quite a few places where "glacier" would be more appropriate, e.g. L22, L25, L26, L284.
  *Changed.*
- L25: "clearest indicators of the climate system's response to recent global warming". This is an odd sentence because warming is part of the climate – rephrase.
  *"climate system" changed to "cryosphere".*
- L26: "Photographic and satellite imagery of reductions…". This could be phrased better, e.g., "photographs and satellite imagery of glacier extents from the past century document widespread retreat".
  *Changed.*
- L35: "the Holocene *was* considerably more stable"
  *Changed.*
- L36: "suggests other mechanisms *controlled glacier size* besides NH insolation"
  *Updated.*
- L57: "Beartooths" – this is colloquial, change to Beartooth Mountains.
  *Changed.*
- L62: Change "course" to "coarse-crystalline"
  *Changed.*
- L298-302: this sentence does not work grammatically.
  *Updated: "We hypothesize that retreat of the Rock Creek glacier from its LGM limit began later in the window of LGM retreat (22 to 18 ka) similar to the Pine Creek moraine (18.2 ± 1.3 ka). After which the Rock Creek glacier experienced slower retreat similar to that of the Clarks Fork glacier after abandoning the Crandall Creek moraine (18.2 ± 0.8 ka), thus allowing for the formation of morainal features within the Lower Valley. Eventually the glacier left the Lower Valley in response to rising atmospheric greenhouse gases."*
- L318: Write out "Younger Dryas" before using the abbreviation.
  *Changed.*
- L376 – grammar needs checking.
  *Changed "which form" to "and formed".*
- L398: highlight*ed* - past tense
  *Changed.*
- L403-404: "Similar influence of topography have been shown to influence" – rewrite this.
  *Changed to: "Similar topographic scenarios have been shown to influence alpine glacier advances…"*

The figures require some improvements:

- 1 is missing lat/long coordinates and a legend for the colour scale. It would be more helpful for the inset map to show a box or star locating the study area specifically, rather than just colouring in the two states. I recommend making the solid-black moraine lines a bit bolder so that they stand out.
  *Lat/long coordinates are added along with a color scale for elevation. Inset map includes a symbol indicating the location of the study area. Moraine lines are more visually apparent.*
- 2 also needs lat/long coordinates. I suggest outlining the white boulder dots, and also enlarging this figure so it takes up the full page width.
  *Lat/long coordinates are now included along with bolder lines around the sample symbols. Figure is now larger.*

Line comments:

L28-29. I disagree that analysis of glacier sensitivity to climate change is limited by the instrumental record, which only goes back decades. Many studies have dated ancient moraines - as this one does – to constrain glacier sensitivity to climate changes of much larger magnitudes than those provided by instrumental records.

*Sentence updated to clarify the disadvantages of a temporally-limited instrumental record: "However, analysis of glacier sensitivity to climate change using the instrumental record is limited by data which only does back decades."*

L218. The text refers the reader to Table 1 for the ages, but the ages are missing in Table 1. Figure 1 doesn't help either because the ages reported there aren't given sample names, so they can't be cross-referenced. Same issue with L 229-230 and elsewhere. Add the ages to Table 1.

*Ages now included in Table 1.*

L231-239. This is all interpretation and discussion, which shouldn't really be mixed with the Results. Move to the Discussion. I would also question whether a population of only four ages can be interpreted as having a multi-modal distribution.

*These lines are now moved to the discussion section for the Neoglacial.*

L366. I think this should refer to Fig. 5, as insolation is not shown in Fig. 6.

*Reference to Fig. 6 is no longer included in this sentence.*

L383. Figure 6, not 5?

*Corrected.*

L406-407. This would place deglaciation within historical records between about 1647 and 1899. Are there any historical accounts of this?

*Unfortunately historical records for this region are limited or non-existent.*

---

## Author Comment (AC2)

This manuscript presents an intriguing set of exposure ages from late Pleistocene to Holocene glacial deposits in the Beartooth Mountains of the western United States. The late Pleistocene ages of the oldest dated moraine position and potential associations with the Younger Dryas will attract a lot of interest. But in my view, the most newsworthy aspect of the study is that it provides rare age control for the late Holocene record of glaciation in this region, which remains sparsely documented. The exposure ages are rather scattered on the youngest landforms, but despite this scatter, the data provide a clear signature of glacial activity during the Neoglacial and LIA.

*Thank you for the helpful assessment and review of our paper. We agree that the late-Holocene ages are noteworthy and provide valuable insight into Neoglacial and LIA glaciation in this region.*

Regarding the youngest landforms, there's some acknowledgement of the complexities involved in interpreting these ages as either climate-driven glacial events or periglacial processes leading to rock glacier formation. I would like to see more discussion about that complex issue and the caution in interpreting such ages.

*Agreed. We will expand upon possible interpretations of the late-Holocene data – particularly related to the possible influence of rockfall and rock glaciation. The additional explanation should acknowledge limitations of the data while supporting the conclusions presented in this paper.*

Overall, however, the ages reported in this manuscript are valuable and should be published because they advance our knowledge of late Pleistocene glacial events and late Holocene stabilizations/readvances of mountain glaciers in the western US.

I have a list of more detailed suggestions below, and recommend publication after moderate revisions.

Line 60: correct spelling is Absaroka

*Corrected.*

Line 63: Replace Course with Coarse

*Corrected.*

Figure 1: A more specific locator inset map with the Beartooth Mountains labeled would be more helpful than highlighting the entire states of Montana and Wyoming. Main map needs lat-lon tics.

*Agreed. We are adding a symbol to indicate the study are location along the border of Montana and Wyoming. Latitude and longitude ticks are to be added as well.*

Figure 2: This figure would be more informative if additional information was added, such as other glacial deposits or features, former ice flow directions, etc. There's also a lot of area included on eastern side of the map that doesn't include any plotted moraine or age information and therefore doesn't need to be shown.

*Figure 2 is to be cropped, thus eliminating the excess eastern portion of the map and focusing more on the glacial deposits discussed in this paper. We will add generalized ice-flow directions and map the rock glacier located to the west of Triangle Lake. Latitude and longitude ticks are to added on this map as well.*

Figure 2 again: With this degree of scatter among the exposure ages, it may not be meaningful to report an average age. Instead, reporting the range of ages could be more useful for interpretations. The scatter also makes me wonder if these features are rock glaciers or protalus ramparts, rather than moraines? This is acknowledged later in the discussion, but a satellite image base map would help visualize the landforms at these field sites.

*Figure 2 will now include satellite imagery as a base map to assist with visualization of the landforms and provide support for our interpretation as moraines – based on morphology and geometry. We plan to remove the mean ages listed beneath each distribution and will discuss the range of ages in the text.*

Line 118: What exactly is meant by "deflated" moraine? Does this refer to ice-stagnation features or melting of an ice-cored moraine? Or wind deflation of fine-grained material from the surface?

*We plan to expand upon this point by discussing the effects of wind deflation of fine-grained material and settling of sediments within the moraine. We will reference recent work by Sortor (2022) where cosmogenic nuclide measurements were used to estimate the rate of moraine deflation within the Front Range of Colorado, USA, which is analogous to the moraines of our study area.*

Line 175: I'm not sure why it's useful to mention how many samples were collected if they weren't all measured.

*Agreed. The line is removed and now reads: "Eighteen samples were collected and processed for $^{10}$Be extraction that best represent our selection criteria…".*

Results section: Ages reported in this section do not match the ages plotted on Figure 2. This needs to be corrected.

*Thank you for identifying this error. All ages in the figures and manuscript are now reported using the Promontory Point production rate and LSDn scaling scheme.*

Line 226: It does not seem warranted to report these exposure ages to the nearest year. I suggest rounding to the nearest decade or century, or whatever is justified by the actual precision of the results. Same comment for other quoted ages throughout the manuscript.

*Agreed. Late-Holocene ages and uncertainties are rounded to the nearest decade based on the average decadal uncertainty for those ages, while Pleistocene ages and uncertainties are rounded to the nearest century for the same reason.*

Lines 231-234: I'm not following the logic here. If rockfall is suspected, then the youngest ages may be reflecting the timing of that rockfall rather than a period of glaciation.

*Ages for the youngest moraines will now be presented as ranges instead of a mean age or the youngest age. We will interpret the ages as a limiting range of deglaciation from the moraine and suggestive of glacial conditions prior to abandonment.*

Lines 238-239: Again, considering the youngest ages to be the minimum-limiting ages of glaciation is risky if rockfall delivery is suspected.

*Same as above.*

Lines 244-245: What about choice of scaling? How much does that change the ages? I don't see Table 2 in the main text.

*Discussion of other production rates have been removed to reduce confusion described by Reviewer #1. As such, we are not inclined to discuss the effect of various scaling schemes on our reported ages at this time and Table 2 is also no longer necessary.*

Lines 338-341: The apparent variability in the timing of moraine abandonment in the YD stadial seems to fall within age uncertainties, so it's difficult to make a strong case for this.

*Agreed. We added the following sentence at the end of the paragraph to emphasize this point: "However, multi-centennial scale uncertainties within each age prevent conclusive attribution of moraines to early or later periods of the YD."*